# The Unfolded Protein Response: An Overview

**DOI:** 10.3390/biology10050384

**Published:** 2021-04-29

**Authors:** Adam Read, Martin Schröder

**Affiliations:** 1Department of Biosciences, Durham University, South Road, Durham DH1 3LE, UK; adam.read2@durham.ac.uk; 2Biophysical Sciences Institute, Durham University, South Road, Durham DH1 3LE, UK

**Keywords:** UPR, IRE1, PERK, ATF6, RIDD, ERAD, inactivation

## Abstract

**Simple Summary:**

The unfolded protein response (UPR) is the cells’ way of maintaining the balance of protein folding in the endoplasmic reticulum, which is the section of the cell designated for folding proteins with specific destinations such as other organelles or to be secreted by the cell. The UPR is activated when unfolded proteins accumulate in the endoplasmic reticulum. This accumulation puts a greater load on the molecules in charge of folding the proteins, and therefore the UPR works to balance this by lowering the number of unfolded proteins present in the cell. This is done in multiple ways, such as lowering the number of proteins that need to be folded; increasing the folding ability of the endoplasmic reticulum and by removing some of the unfolded proteins which take longer to fold. If the UPR is successful at reducing the number of unfolded proteins, the UPR is inactivated and the cells protein folding balance is returned to normal. However, if the UPR is unsuccessful, then this can lead to cell death.

**Abstract:**

The unfolded protein response is the mechanism by which cells control endoplasmic reticulum (ER) protein homeostasis. Under normal conditions, the UPR is not activated; however, under certain stresses, such as hypoxia or altered glycosylation, the UPR can be activated due to an accumulation of unfolded proteins. The activation of the UPR involves three signaling pathways, IRE1, PERK and ATF6, which all play vital roles in returning protein homeostasis to levels seen in non-stressed cells. IRE1 is the best studied of the three pathways, as it is the only pathway present in *Saccharomyces cerevisiae*. This pathway involves spliceosome independent splicing of *HAC1* or *XBP1* in yeast and mammalians cells, respectively. PERK limits protein synthesis, therefore reducing the number of new proteins requiring folding. ATF6 is translocated and proteolytically cleaved, releasing a NH_2_ domain fragment which is transported to the nucleus and which affects gene expression. If the UPR is unsuccessful at reducing the load of unfolded proteins in the ER and the UPR signals remain activated, this can lead to programmed cell death.

## 1. Introduction

The endoplasmic reticulum (ER) is a membrane-bound organelle that is responsible for folding, modification and synthesis of secretory and organelle-bound proteins. The process of protein synthesis and folding is highly controlled and is sensitive to perturbation of ER homeostasis. This situation is often referred to as ER stress. A multitude of homeostatic changes can lead to a build-up of unfolded proteins. The changes in ER homeostasis can be caused by Ca^2+^ depletion, hypoxia, altered glycosylation or viral infection [1,2,3]. Consequently, because of the accumulation of unfolded proteins, the cell has evolved a response mechanism to prevent further accumulation of unfolded proteins. This signalling pathway is known as the unfolded protein response (UPR). The UPR detects the build-up of unfolded or misfolded proteins and adjust the protein folding ability of the ER.

The UPR in mammalian cells is complex and works via three principal ER transmembrane receptors: type I transmembrane protein inositol requiring 1 (IRE1α); eukaryotic initiation factor 2α (eIF2α) kinase (PERK) and activating transcription factor 6 (ATF6) [1,4,5,6]. The UPR attempts to process the accumulation of unfolded proteins by downregulating transcription of secretory proteins and increases the removal of misfolded or slowly folding proteins through ER-associated degradation (ERAD) or lysosomal degradation [7]. Additionally, the synthesis of ER resident chaperones and foldases is increased to promote the folding ability of the ER and to alleviate the unfolded protein burden [8,9]. If adaptation by the cell via the UPR is not sufficient to deal with the increased load of unfolded proteins, activation of JNK protein kinase and caspases 3, 7 and 12 occurs, which ultimately leads to an apoptotic response [6,10].

IRE1α is a type I transmembrane protein with a protein serine/threonine kinase and endoribonuclease domain. Activation of IRE1α occurs when the ER chaperone BiP is released from its luminal domain when a build-up of unfolded proteins arises [1,5,11]. Once IRE1α is activated, its endoribonuclease domain initiates splicing, independent of the spliceosome, of the mRNA encoding the bZIP transcription factor XBP-1 in mammals and Hac1 in yeast by its ortologue Ire1 [6,12]. PERK is also a type I transmembrane protein which shares luminal domain homology with IRE1α [13]. When PERK is activated, it phosphorylates the eukaryotic translation initiation factor 2α (eIF2α) at serine 51 [14]. Protein synthesis is inhibited by serine 51 phosphorylated eIF2α, which in turn reduces the number of proteins that need to be folded. *ATF6* is a type II transmembrane protein. When unfolded proteins accumulate, ATF6 is transported to the Golgi complex where it is proteolytically cleaved by S1P and S2P to release the NH_2_ terminal-domain [15,16].The cleaved section of ATF6 translocates to the nucleus, where it activates gene transcription of target genes such as *GRP78*, *GRP94* and *calnexin* [17]. The ER-resident proteins that are regulated by the UPR share a common activating sequence termed the unfolded protein response element (UPRE) or ER stress response elements (ERSEs). These elements are necessary for activation of transcription in response to accumulation of unfolded proteins [18,19].

## 2. Ire1 in Yeast

The UPR in yeast differs from mammals, as it lacks both ATF6 and PERK, which leaves only one known pathway originating at the IRE1α orthologue Ire1 available to yeast cells to respond to accumulation of unfolded proteins in the ER. There are two proposed models for the activation of the UPR via Ire1. One model suggests direct binding of unfolded proteins to Ire1, which causes oligomerisation of Ire1. This is suggested by studies of the core of Ire1′s luminal domain, which contains two interfaces [20]. Interface 1 creates a deep groove. Interface 2 allows oligomerisation. Mutating either of the interfaces decreases splicing of the *HAC1* mRNA by Ire1 and decreases oligomerisation of Ire1 [21]. The grooves formed by Ire1 are similar to that of the major histocompatibility complex (MHC), which are able to bind peptides with high specificity. The Ire1 groove is lined with hydrophobic and hydrophilic residues [20]. It is also thought that due to the depth of the groove, correctly folded proteins are unable to access the groove and therefore only allow Ire1 to bind to unfolded proteins. This steric discrimination may be the reason for the specificity of Ire1 for unfolded proteins [20]. The other proposed model involves the resident ER chaperone Kar2, called BiP in mammalian cells, which is bound to inactivate Ire1 and preferentially binds to unfolded proteins when they accumulate. This releases Kar2 from Ire1 allowing it to oligomerise, which promotes *trans*-autophosphorylation of Ire1 in its activation segment at serines 837, 840, 841, 850, and threonine 844 [22,23,24] and induction of endoribonuclease activity [1]. The target of Ire1 endoribonuclease activity is a mRNA that encodes the transcription factor Hac1 which binds to the UPRE in the promoters of target genes such as *KAR2* [25,26]. During activation of the UPR, an intron of 252-nucleotides is removed from the *HAC1* mRNA creating the spliced form of *HAC1,* termed *HAC1^i^* (Figure 1) [19]. The next step after cleavage of *HAC1* mRNA by Ire1 is the ligation of the two exons by tRNA ligase. The splicing of *HAC1* is unique, as most other pre-mRNAs require the spliceosome and its related constituents. By contrast, the type of splicing for *HAC1* is more like pre-tRNA splicing, as it only requires two components for cleavage and ligation [27]. The newly ligated mRNA is then translated to an active protein able to bind to the promoter regions of target genes (Figure 1).

The UPR varies even among different yeast species where the Ire1 pathway is the only conserved pathway. The main difference is that some species focus on processing of *HAC1* mRNA, such as *Saccharomyces cerevisiae* and *Candida albicans* [28], whereas some species, such as *Schizosaccharomyces pombe*, focus on the degradation of mRNAs which have been localised to the ER through a mechanism known as RIDD [28].

## 3. IRE1 and RIDD

The RNase activity of metazoan IRE1 is also involved in a different mechanism known as regulated IRE1-dependent decay (RIDD). RIDD was reported to degrade mRNAs that are localised to the ER [29]. The cleavage site for the mRNAs seems to be similar to the cleavage sequence of *XBP1*, and the mRNA fragments are then degraded by cellular exoribonucleases [30]. IRE1 has two different isoforms, IRE1α and IRE1β, both of which have the ability to splice *XBP1* and to activate RIDD, although IREα shows stronger splicing activity and IRE1β exerts stronger RIDD activation [31]. RIDD has not been observed in *S. cerevisiae*, as it appears that *HAC1* splicing is the only target for RNase activity, whereas in *S. pombe* the splicing of *HAC1* does not occur and RIDD is the main mechanism for ER protein homeostasis [32]. It has also been demonstrated that basal levels of RIDD are required to maintain ER homeostasis. As *XBP1*/*HAC1* is only spliced under elevated ER stress, basal activity of RIDD is observed as the cleavage of mRNAs when cells are only under basal levels of ER stress. IRE1β has many target substrates even when ER stress is not induced, further supporting the theory of basal RIDD activity [33]. *ire1α* and *xbp1* mutants were also used to investigate which one had the greater effect on ER homeostasis when not under stress and the *ire1α* mutant altered homeostasis to a greater extent [30,33]. This suggests that IRE1α has a way of balancing ER homeostasis even when the cell is not under stress through mechanisms such as RIDD. Under normal unstressed cells, *XBP1/HAC1* is not spliced. Once ER stress due to the accumulation of unfolded proteins occurs, *XBP1/HAC1* splicing is present and increases until it reaches a maximum peak and then starts to decrease. RIDD is active under basal conditions and once ER stress is induced also increases. If both splicing and RIDD are able to alleviate the unfolded protein load in the ER, then homeostasis returns to normal levels [34,35]. However, if the mechanisms of *XBP1/HAC1* splicing and RIDD are insufficient to normalise ER homeostasis, then, unlike splicing, which decreases over prolonged ER stress, RIDD continues increasing and degrades pre-miRNAs as well as pro-survival protein encoding mRNAs [36,37]. This ultimately leading to apoptosis of the cell and is termed prodeath RIDD [30].

## 4. IRE1 Activates JNK Signaling

Activated IRE1α has also been revealed to recruit TNF receptor-associated factor 2 (TRAF2), the newly recruited TRAF2 then also recruits apoptosis signal-regulating kinase (ASK1) which directly interacts with TRAF2 [38]. Once ASK1 is recruited and activated, a signal is relayed to c-Jun amino-terminal kinase (JNK) and p38 [10]. This apoptotic pathway of IRE1α further demonstrates its ability to control the fate of the cell under ER stress. 

## 5. Ire1 Inactivation

The mechanism for the inactivation of Ire1, the most studied of all the UPR pathways, has yet to be fully elucidated. However, two phosphatases Dcr2 and Ptc2 have been suggested to negatively regulate yeast Ire1 [39,40,41,42,43]. Reversible protein phosphorylation is an important mechanism of control of protein activity which works through the antagonistic actions of protein kinases and phosphatases. Ptc2 is a protein serine/threonine phosphatase which dephosphorylates Ire1 through a direct interaction with Ire1 [39]. Ptc2 binds to Ire1 in vitro and has specificity for phosphorylated Ire1. Binding to Ire1 is independent of divalent metal ions or mutations in the metal ion binding site of Ptc2. Ptc2 dephosphorylates Ire1 in a Mg^2+^ or Mn^2+^ dependent manner [39]. The dephosphorylation inactivates Ire1 and prevents splicing of *HAC1*, which attenuates the UPR. Cells lacking *PTC2* have shown similar phenotypes to cells overexpressing Ire1 which showed a three-fold increase in *HAC1* splicing. Additionally, when catalytically inactive Ptc2 is expressed, a similar increase in *HAC1* splicing was observed which was not seen with active Ptc2 overexpression [40,42]. However, *PTC2* is not necessary for cell survival during ER stress, as *PTC2* null mutants still showed the same growth as wild-type cells, even though they had an increase in UPR activity [22,39]. This may suggest that additional phosphatases compensate for the loss of *PTC2*. *PTC2* is also constitutively expressed and is not regulated by Ire1 or induced upon activation of the UPR. Therefore, the exact mechanism of regulation of Ire1 by Ptc2 remains unclear [39].

Dose-dependent cell cycle regulator (Dcr2) has also been suggested to negatively regulate the UPR at a step preceding *HAC1* splicing [42]. Mutants with Ire1-S840E-S841E or Ire1-S840A-S841A as their sole version of Ire1 represent phosphorylated and non-phosphorylated versions of Ire1, respectively [42]. Dcr2 directly interacted with the phosphomimic Ire1 mutant but not the non-phosphorylatable mutant suggesting that Dcr2 shows specificity towards phosphorylated Ire1 [42]. However, loss of *DCR2* has no impact on cell survival of ER stress suggesting that Dcr2 is not the main mechanism for Ire1 dephosphorylation [22,42]. In addition, survival of ER stress was not affected by loss of both Dcr2 and Ptc2 [22], suggesting that other phosphatases are involved in inactivating Ire1, that other phosphatases can compensate for the loss of both Dcr2 and Ptc2, or that dephosphorylation is not necessary for inactivation of Ire1. The kinetics of inactivation of an Ire1 mutant lacking all phosphorylation sites in its activation segment was indistinguishable from wild type Ire1 [22]. These data provide evidence for dephosphorylation-independent inactivation of Ire1 or biphasic inactivation of Ire1, where fast dephosphorylation steps precede slower dephosphorylation-independent inactivation steps.

## 6. PERK Activation and Signalling

The luminal domains of IRE1 and PERK show a small amount of homology in their luminal domains and both IRE1 and PERK from *Caenorhabditis elegans* can function as replacements for the luminal domain in *S. cerevisiae*, even though yeast does not have the *PERK* gene [13,44]. The activation of PERK is also thought to be similar to that of IRE1, as both rely on BiP bound to their luminal domains being released due to its affinity for unfolded proteins, which is followed by oligomerisation and *trans*-phosphorylation of PERK monomers [44], although the way in which BiP represses the activation of IRE1 and PERK may differ slightly. The BiP binding sequence in IRE1 overlaps with the region believed to be involved in oligomerisation and signalling [45], whereas in PERK, the BiP binding region does not overlap with the oligomerisation sequence and, therefore, is thought to interfere with PERK activation sterically [13,45]. Once activated PERK phosphorylates eukaryotic initiation factor 2 on its α subunit (eIF2α), Figure 2 [13]. eIF2α binds to methionylated initiator methionyl-tRNA, GTP and the 40 S ribosomal subunit to form the 43 S preinitiation complex [44]. This preinitiation complex binds to the 5′ cap structure of mRNAs and in a 5′ to 3′ manner scans the mRNA until it reaches the first AUG codon, which then allows binding of the 60 S subunit to initiate translation [44]. When the two ribosomal subunits combine, the GTP in the preinitiation complex is hydrolysed to GDP. Phosphorylated eIF2 reduces the exchange of eIF2-GDP to eIF2-GTP which in turn reduces translation of upstream open reading frames (ORFs) which would normally be translated when GTP is readily available [46]. High levels of phosphorylated eIF2α lead to downstream ORFs which would not be translated in normal circumstances due to the ribosomes still being bound to the mRNAs but scanning further along the mRNAs before re-initiation of translation is started due to lower amounts of GTP [44,46,47]. This mechanism represses translation of most mRNAs except for *GCN4* in yeast and *ATF4* in vertebrates, which are actually increased (Figure 2) [47,48]. ATF4 has been demonstrated to induce expression of both ATF3 and CHOP under ER stress (Figure 2). CHOP is a vital intermediary of ER stress-induced apoptosis as it induces multiple pro-apoptotic molecules such as death receptor 5 (DR5) and tribbles homologue 3 (TRB3) [49]. ATF3 is also an important molecule induced by ATF4, as ATF3 and ATF4 have been suggested to bind to the promoter region of *GADD34.* GADD34 is a regulatory subunit of protein phosphatase 1, which targets protein phosphatase 1 to phosphorylated eIF2α, and in this way promotes dephosphorylation of eIF2α (Figure 2) [49,50,51,52,53].

## 7. ATF6 Signalling

ATF6 is a type II transmembrane protein and the third pathway which plays a vital role in UPR signalling. Similarly to IRE1 and PERK, this transmembrane protein is also bound to BiP in its inactive state and requires dissociation of BiP before the signalling cascade can activate [44]. Regulated intermembrane proteolysis (RIP) is the process by which transmembrane proteins such as ATF6 are cleaved to release their cytoplasmic domains which then regulate gene expression by entering the nucleus [15,52]. The mechanism for the activation of ATF6 is best represented by the model for sterol regulatory element binding proteins (SREBPs), which are transcription factors important in the synthesis of cholesterol [53]. The SREBPs are also embedded in the ER membrane with their NH_2_-terminal and COOH-terminal segments projecting into the cytosol and a hydrophilic loop facing the ER lumen which is similar to ATF6 as its NH_2_ domain also projects into the cytosol [15]. The next step for SREBPs is to bind with SREBP cleavage activating-protein (SCAP), which transports the SREBPs to the Golgi complex, where RIP occurs [54,55]. Site-1 protease (S1P) is a membrane-anchored serine protease that cleaves the recognition sequence RXXL, where X can be any amino acid [15,56]. S1P cleavage splits the two transmembrane domains, with the NH_2_-terminal segment being termed as the intermediary fragment (Figure 3) [15]. Site-2 protease (S2P) is responsible for cleaving the intermediary fragment, which releases the NH_2_-terminal fragment that then localises to the nucleus and binds to promotor regions initiating gene expression (Figure 3) [57,58]. The same RXXL sequence has been detected in the luminal domain of ATF6 which suggests that S1P may also cleave ATF6 in a similar manner to SREBP-2 [15,17]. However, an equivalent molecule to SCAP has not been identified for ATF6. Therefore, the transport of ATF6 to where S1P is localised, which is usually in or near the Golgi complex, remains unknown [15].

## 8. ERAD Induction

ER-associated degradation (ERAD) is a mechanism by which slowly folding or misfolded proteins are cleared from the ER and degraded [8]. The clearance and degradation of these unfolded or misfolded protein helps to alleviate the accumulation of unfolded proteins during ER stress. The UPR and ERAD are known to functionally connect to each other. Cells with compromised ERAD display constant UPR activation due to misfolded proteins not being degraded [43,59,60]. In mammalian cells, ATF4 and ATF6 have been suggested to induce HERP/Mif1, an ER membrane protein which is known to interact with the 26 S proteosome, which brings proteosomes closer to the ER allowing for efficient ERAD [43]. Additionally, *herp*^-/-^ cells have shown greater susceptibility to ER stress and demonstrated increased UPR signaling as well as stabilisation of an endogenous ERAD substrate [61]. Multiple genes involved in ERAD, such as EDEM, HRD1 and UGGT, require induction by XBP1. EDEM recognises and targets unfolded proteins for degradation [62]. It has also been suggested that the UPR may work in two phases. Phase one allows the unfolded proteins time to refold without being degraded and phase two degrades any proteins which have failed to fold [63]. This is believed to occur due to ATF6 activating quickly when compared to IRE1, which has to induce *XBP1* splicing and translation of *XBP1* to an active protein. During this time, the ER chaperones induced by ATF6 are able to promote protein folding before spliced XBP1 induces ERAD genes which promote the degradation of unfolded proteins [63,64].

## 9. Future Perspectives

The importance of protein folding homeostasis in the ER has been demonstrated to have a vital role in the function and growth, as well as being a major regulator at transcriptional, translational and post-translational levels. The basic mechanisms of action have been established for IRE1, PERK and ATF6. However, a deeper understanding of the interactions of these signaling pathways with molecular chaperones is needed. There are still important questions left unanswered. Are all UPR pathways regulated in the same manner? Is BiP the only molecular chaperone involved at the start of all three UPR pathways, or has the importance of BiP been overstated and do other molecular chaperones contribute to the regulation of the ER stress sensors? The factors that regulate the UPR pathways still remain uncertain. The characterisation of these factors could lead to a greater understanding of the effect that the UPR has in the cell. For example, the UPR is not only focused on clearing accumulated unfolded proteins, but it also has the ability to initiate or prevent apoptosis. Therefore, understanding all the factors involved in the UPR could demonstrate the ability of the UPR to affect previously unrelated pathways.

## Figures and Tables

**Figure 1 biology-10-00384-f001:**
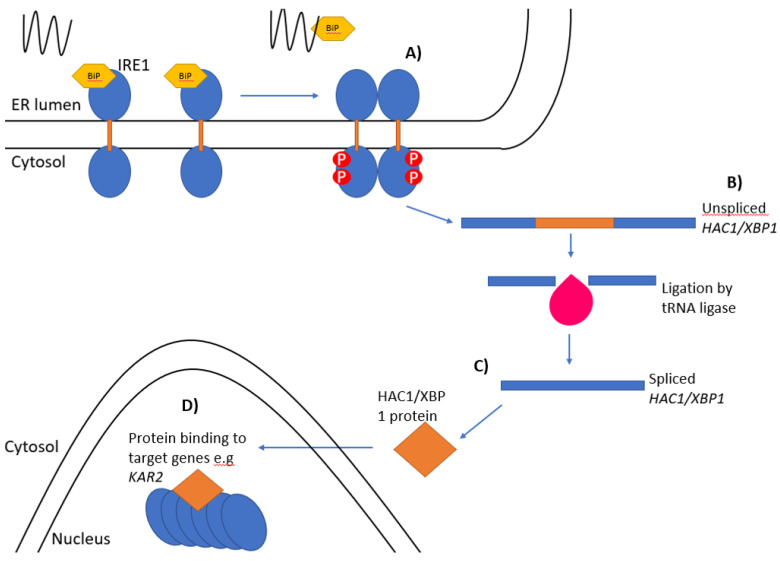
The unfolded protein response. The Ire1 pathway of the unfolded protein response: (**A**) Ire1-bound BiP/Kar2 (yellow) is released from the luminal domain of Ire1 when unfolded proteins accumulate leading to the oligomerisation and *trans*-autophosphorylation of Ire1 monomers. (**B**) Unspliced *HAC1/XBP1* mRNA is spliced by the endoribonuclease domain of Ire1 and the two exons ligated together by tRNA ligase (pink). (**C**) The *HAC1^i^/XBP1^s^* mRNA is then translated into an active protein. (**D**) The newly synthesised protein then enters the nucleus and binds to the promoter regions of target genes.

**Figure 2 biology-10-00384-f002:**
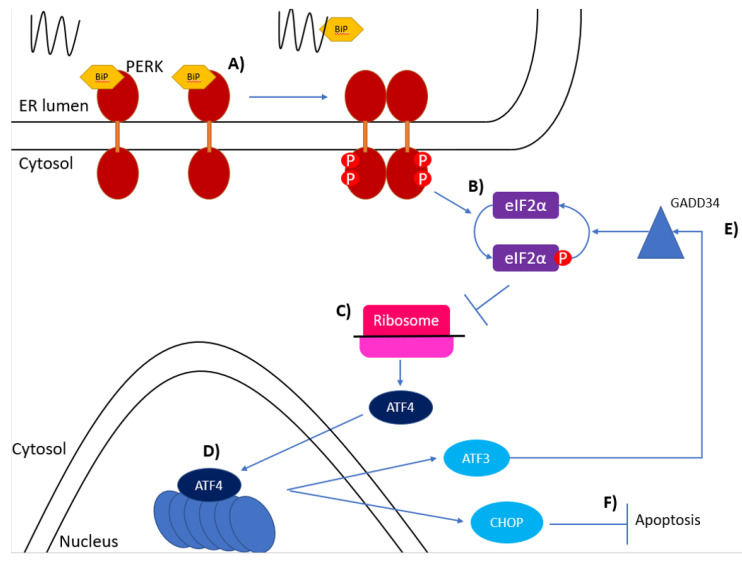
The PERK pathway of the unfolded protein response. (**A**) BiP bound to the luminal domain of PERK dissociates when unfolded proteins accumulate allowing oligomerisation and phosphorylation of the PERK monomers. (**B**) Phosphorylated PERK then phosphorylates eIF2 on its α subunit at S51. (**C**) Phosphorylated eIF2α interferes with ribosomal translation of mRNA and causes downstream ORFs to be translated, such as ATF4. (**D**) ATF4 induces expression of ATF3 and CHOP. (**E**) ATF3 binds to the promoter region of *GADD34* which activates a protein phosphatase which dephosphorylates eIF2α. (**F**) CHOP induces ER stress-induced apoptosis.

**Figure 3 biology-10-00384-f003:**
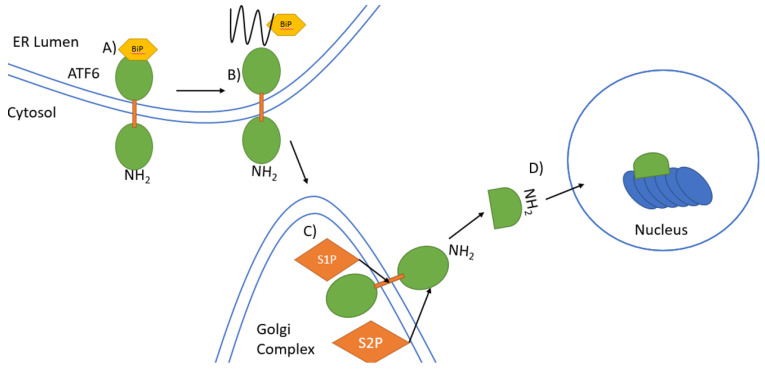
Signal transduction by ATF6 in the UPR. (**A**) In unstressed conditions, binding of BiP to the COOH terminal ER luminal domain of the type II transmembrane protein ATF6 keeps ATF6 inactive. The NH_2_ terminal domain is located in the cytosol. (**B**) Under ER stress unfolded proteins accumulate leading to dissociation of BiP from ATF6. ATF6 then translocates to the Golgi complex from the ER. (**C**) ATF6 is first cleaved by S1P in the transmembrane domains and then by S2P in the cytosolic domain near the ER membrane to release the NH_2_-terminal domain. (**D**) The NH_2_-terminal domain migrates to the nucleus where it induces gene expression.

## Data Availability

Not applicable.

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
