# Peer review of "The Unfolded Protein Response: An Overview"

_biology, 2021, doi:10.3390/biology10050384_

Round 1
Reviewer 1 Report
This manuscript reviews a central topic in cell physiology which is always of high interest. It is well written and covers the main aspects involved in ER stress regulation and UPR pathways.
Because the paper includes studies made in different model organisms, is not easy to follow which species the authors are talking about. Authors should help the reader by including as possible the cellular type in which the referred findings were described.
It would be illustrative to include a figure of the PERK pathway to cover the main mammalian UPR pathways.
It is important that authors include a paragraph in the second chapter indicating that Ire1 mechanisms differ between yeast species as reviewed in Hernández-Elvira et al, Cells 7:106 (2018)
The 8 chapter (ERAD induction) should include some recent references since it is based on only four different references, here I include some examples:
Hwang and Qi (2018) Quality Control in the Endoplasmic Reticulum: Crosstalk between ERAD and UPR pathways. Trends in Biochemical Sciences 43,593
Shuangchan et al, (2020) The Integrated UPR and ERAD in Oligodendrocytes Maintain Myelin Thickness in Adults by Regulating Myelin Protein Translation. Journal of Neuroscience 40 (43) 8214-8232
Qu et al, (2021) The Roles of the Ubiquitin–Proteasome System in the Endoplasmic Reticulum Stress Pathway. Int. J. Mol. Sci. 22, 1526.
Author Response
Responses to the comments by reviewer 1
We thank the reviewer for her/his comments on our manuscript. We have improved our manuscript along the lines suggested by the reviewer. The changes in response to the reviewer’s recommendations are described in detail below.
This manuscript reviews a central topic in cell physiology which is always of high interest. It is well written and covers the main aspects involved in ER stress regulation and UPR pathways.
Because the paper includes studies made in different model organisms, is not easy to follow which species the authors are talking about. Authors should help the reader by including as possible the cellular type in which the referred findings were described.
The PERK and ATF6 pathways are only known to be present in metazoans, not in yeast species. Therefore, we believe that the model system is clear when discussing these two pathways.
The IRE1 pathway is present in yeast and higher eukaryotes. We distinguish between yeast and mammalian systems by referring to the yeast orthologue of IRE1 as Ire1 (in accordance to nomenclature conventions for S. cerevisiae) and metazoan/mammalian IRE1 as IRE1α or IRE1β (in accordance with nomenclature conventions for mammalian genes and gene products). IRE1 is used to refer to several orthologues/isoforms of IRE1 at the same time, or to IRE1 in metazoan species that only possess one IRE1 gene, for example C. elegans.
We have added extra information at lines 61 and 76 of the revised manuscript to introduce Ire1 and the yeast orthologue of mammalian IRE1.
It would be illustrative to include a figure of the PERK pathway to cover the main mammalian UPR pathways.
An additional figure 2 depicting the PERK pathway has been added. The numbering of all subsequent figures has been changed.
It is important that authors include a paragraph in the second chapter indicating that Ire1 mechanisms differ between yeast species as reviewed in Hernández-Elvira et al, Cells 7:106 (2018)
A paragraph indicating that the mechanisms of UYPR signalling differ between yeast species has been inserted (lines 111 – 115) in the revised manuscript.
The 8 chapter (ERAD induction) should include some recent references since it is based on only four different references, here I include some examples:
Hwang and Qi (2018) Quality Control in the Endoplasmic Reticulum: Crosstalk between ERAD and UPR pathways. Trends in Biochemical Sciences 43,593
Shuangchan et al, (2020) The Integrated UPR and ERAD in Oligodendrocytes Maintain Myelin Thickness in Adults by Regulating Myelin Protein Translation. Journal of Neuroscience 40 (43) 8214-8232
Qu et al, (2021) The Roles of the Ubiquitin–Proteasome System in the Endoplasmic Reticulum Stress Pathway. Int. J. Mol. Sci. 22, 1526.
Additional and more recent references have been added to this section.
Reviewer 2:
In their manuscript, Read and Schröder give an overview of the current knowledge regarding the unfolded protein response (UPR). After an introduction, they focus their literature review on the 3 signalling pathways involved in UPR: IRE1, PERK and ATF6. The activation and signalisation of those pathways are well presented. In their last section, they focus on the ERAD system, which is interconnected to UPR.
The review is well written and is informative. A couple of minor comments remain:
- As for IRE1 and ATG6, it could be beneficial to have a figure related to PERK signal transduction.
An additional figure 2 depicting the PERK pathway has been added. The numbering of all subsequent figures has been changed.
- Section 7 “IRE1 inactivation” may come after section 4, so all the sections related to IRE1 are one after another.
Section 7 has been moved so that all the sections relating to IRE1 are grouped together and the review flows better.
- For the figures, the authors used the same colours for both IRE1 and ATF6. Even though those figures are clearly labelled for their respective pathway, it might be better suited to use a different colour coding for IRE1 and ATF6.
The three proteins are now shown in different colours in all figures. IRE1 is shown in blue, PERK in red, and ATF6 in green.
- Figure 1: IRE1 should be labelled on the figure (as it is for ATF6).
IRE1 has been labelled in Figure 1.
- Line 285: [1-61] is unnecessary.
Line has been removed.
Reviewer 2 Report
In their manuscript, Read and Schröder give an overview of the current knowledge regarding the unfolded protein response (UPR). After an introduction, they focus their literature review on the 3 signalling pathways involved in UPR: IRE1, PERK and ATF6. The activation and signalisation of those pathways are well presented. In their last section, they focus on the ERAD system, which is interconnected to UPR.
The review is well written and is informative. A couple of minor comments remain:
- As for IRE1 and ATG6, it could be beneficial to have a figure related to PERK signal transduction.
- Section 7 “IRE1 inactivation” may come after section 4, so all the sections related to IRE1 are one after another.
- For the figures, the authors used the same colours for both IRE1 and ATF6. Even though those figures are clearly labelled for their respective pathway, it might be better suited to use a different colour coding for IRE1 and ATF6.
- Figure 1: IRE1 should be labelled on the figure (as it is for ATF6).
- Line 285: [1-61] is unnecessary.
Author Response
Responses to the comments by reviewer 2
We thank the reviewer for her/his comments on our manuscript. We have improved our manuscript along the lines suggested by the reviewer. The changes in response to the reviewer’s recommendations are described in detail below.
In their manuscript, Read and Schröder give an overview of the current knowledge regarding the unfolded protein response (UPR). After an introduction, they focus their literature review on the 3 signalling pathways involved in UPR: IRE1, PERK and ATF6. The activation and signalisation of those pathways are well presented. In their last section, they focus on the ERAD system, which is interconnected to UPR.
The review is well written and is informative. A couple of minor comments remain:
- As for IRE1 and ATG6, it could be beneficial to have a figure related to PERK signal transduction.
An additional figure 2 depicting the PERK pathway has been added. The numbering of all subsequent figures has been changed.
- Section 7 “IRE1 inactivation” may come after section 4, so all the sections related to IRE1 are one after another.
Section 7 has been moved so that all the sections relating to IRE1 are grouped together and the review flows better.
- For the figures, the authors used the same colours for both IRE1 and ATF6. Even though those figures are clearly labelled for their respective pathway, it might be better suited to use a different colour coding for IRE1 and ATF6.
The three proteins are now shown in different colours in all figures. IRE1 is shown in blue, PERK in red, and ATF6 in green.
- Figure 1: IRE1 should be labelled on the figure (as it is for ATF6).
IRE1 has been labelled in Figure 1.
- Line 285: [1-61] is unnecessary.
Line has been removed.